# Pomegranate Pomace Extract with Antioxidant, Anticancer, Antimicrobial, and Antiviral Activity Enhances the Quality of Strawberry-Yogurt Smoothie

**DOI:** 10.3390/bioengineering9120735

**Published:** 2022-11-28

**Authors:** Nouf H. Alsubhi, Diana A. Al-Quwaie, Ghadeer I. Alrefaei, Mona Alharbi, Najat Binothman, Majidah Aljadani, Safa H. Qahl, Fatima A. Jaber, Mashael Huwaikem, Huda M. Sheikh, Jehan Alrahimi, Ahmed N. Abd Elhafez, Ahmed Saad

**Affiliations:** 1Biological Sciences Department, College of Science & Arts, King Abdulaziz University, Rabigh 21911, Saudi Arabia; 2Department of Biology, College of Science, University of Jeddah, Jeddah 21589, Saudi Arabia; 3Department of Biochemistry, College of Science, King Saud University, Riyadh 11451, Saudi Arabia; 4Department of Chemistry, College of Sciences & Arts, King Abdulaziz University, Rabigh 21911, Saudi Arabia; 5Cinical Nutrition Department, College of Applied Medical Sciences, King Faisal University, Al Ahsa 31982, Saudi Arabia; 6Department of Biological Sciences, Faculty of Sciences, King Abdulaziz University, Jeddah 21589, Saudi Arabia; 7Immunology Unit, King Fahad Medical Research Centre, King Abdulaziz University, Jeddah 21589, Saudi Arabia; 8Department of Internal Medicine, Faculty of Medicine, Zagazig University, Zagazig 44519, Egypt; 9Biochemistry Department, Faculty of Agriculture, Zagazig University, Zagazig 44511, Egypt

**Keywords:** pomegranate, pomace, extraction, biological activity, functional beverages, sensory quality

## Abstract

Valorizing the wastes of the food industry sector as additives in foods and beverages enhances human health and preserves the environment. In this study, pomegranate pomace (PP) was obtained from the company Schweppes and exposed to the production of polyphenols and fiber-enriched fractions, which were subsequently included in a strawberry-yogurt smoothie (SYS). The PP is rich in carbohydrates and fibers and has high water-absorption capacity (WAC) and oil-absorption capacity (OAC) values. The LC/MS phenolic profile of the PP extract indicated that punicalagin (199 g/L) was the main compound, followed by granatin B (60 g/L) and pedunculagin A (52 g/L). Because of the high phenolic content of PP extract, it (*p* ≤ 0.05) has high antioxidant activity with SC50 of 200 µg/mL, besides scavenging 95% of DPPH radicals compared to ascorbic acid (92%); consequently, it reduced lung cancer cell lines’ viability to 86%, and increased caspase-3 activity. Additionally, it inhibited the growth of pathogenic bacteria and fungi i.e., *L. monocytogenes*, *P. aeruginosa*, *K. pneumonia*, *A. niger*, and *C. glabrata*, in the 45–160 µg/mL concentration range while killing the tested isolates with 80–290 µg/mL concentrations. These isolates were selected based on the microbial count of spoiled smoothie samples and were identified at the gene level by 16S rRNA gene sequence analysis. The interaction between Spike and ACE2 was inhibited by 75.6%. The PP extract at four levels (0.4, 0.8, 1.2, and 1.4 mg/mL) was added to strawberry-yogurt smoothie formulations. During 2 months storage at 4 °C, the pH values, vitamin C, and total sugars of all SYS decreased. However, the decreases were gradually mitigated in PP-SYS because of the high phenolic content in the PP extract compared to the control. The PP-SYS3 and PP-SYS4 scored higher in flavor, color, and texture than in other samples. In contrast, acidity, fat, and total soluble solids (TSS) increased at the end of the storage period. High fat and TSS content are observed in PP-SYS because of the high fiber content in PP. The PP extract (1.2 and 1.6 mg/mL) decreases the color differences and reduces harmful microbes in PP-SYS compared to the control. Using pomegranate pomace as a source of polyphenols and fiber in functional foods enhances SYS’s physiochemical and sensory qualities.

## 1. Introduction

Agriculture and the food industry produce a huge amount of waste that poses a threat to health and the environment, although it is rich in bioactive components. Because of the global food shortage, we must reuse the waste in various applications, such as foods and feedstuffs [1,2,3]. Unfortunately, separating bioactive compounds from the consumable parts of the agricultural product could have an excessive cost for the starting material. Accordingly, using agro-industrial waste (such as fruit peels) for separating such compounds to control the spread of foodborne and human pathogens is more economical [4,5]. Pomegranate produces about 1,500,000 tons of waste worldwide per annum [6] and is expected to reach 12,500,000 tons by 2022; about 60% of the fruit weight corresponds to peels, which are often discarded as agro-waste.

Pomegranate (*Punica granatum* L.) has a well-known medicinal history. It is rich in bioactive molecules, phenolic compounds, and flavonoids and has many medicinal properties [7]. The level of phenolic compounds changes according to cultivars and fruit parts [8]. Gözlekçi et al. [9] demonstrated that peel extract contains higher total phenolics, flavonoids, and anthocyanins content than pulp and juice extracts and possesses stronger biological activities. The phytochemical constituents of the ethanolic extract of pomegranate juice were determined by Sorrenti et al. [10], who identified many phytochemical compounds; phenolic compounds were the main constituents. On the other hand, Khaleel et al. [11] demonstrated the presence of 292 different phytocompounds with different retention times and chemical structures. Eleven elements were considered most significant as antimicrobials.

In this regard, Priyanka et al. [12] studied the antimicrobial activity of methanol, chloroform, and methanol chloroform (1:1) extracts of peel and juice of pomegranates in different bacterial and fungal strains. They found that peel methanolic extract showed the highest antimicrobial activity, while chloroform extract was less effective against the tested microorganisms. Other studies have also reported that pomegranate methanol peel extract showed maximum growth inhibition (85.71%) compared to all tested organic solvents against some pathogenic and drug-resistant bacterial strains [13]. Li et al. [14] determined the MICs of punicalagin, as the main active component in pomegranate peel, for ten *Salmonella* strains and found the minimum inhibitory concentrations (MICs) in the range of 0.20–1.0 mg/mL. Moreover, Zam and Khaddour [15] detected a reduction in the adhesion index of up to 80% and a reduction in *E. coli* motility in the urinary tract using aqueous pomegranate peel extract at MIC. Abou El-Nour [16] recorded a large decrease in biofilm formation of *Pseudomonas aeruginosa* and inhibition in swarming motility when treated with aqueous and alcoholic pomegranate peel extracts.

Pomegranate extracts also exhibit antioxidant activity. It has been reported by many investigators that there is a relationship between chemopreventive agents and antioxidant activity; during the oxidation of biological materials, superoxide, H_2_O_2_, and hydroxyl radicals (OH˙) are produced. Reducing the destructive effect on human cells and tissues stops unorganized radical reactions. Mekni et al. [17] illustrated that methanol peel extracts of five pomegranate varieties have high antioxidant power (above 80%) against DPPH radicals. Ohshima et al. [18] demonstrated that oxidative stress generates toxic metabolites which can initiate and stimulate cancers. Breast cancer is the most important life disease a woman may face during her lifetime. Rocha et al. [19] found that the growth of two breast cancer cell lines was completely inhibited by a 1% concentration of pomegranate juice, which also can decrease cell migration to bones. Li et al. [14] recorded a reduction in cell viability in different human cancers by methanol PPE, and MCF-7 breast cells were the most responsive to the antitumor effect (growth inhibition of 81.0% at 5 μg/mL) compared to A549 lung cell cancer (80.0% at dose 250 μg/mL). A recent study by Mahmoudi et al. [20] suggested a decrease in the viability of MCF-7 and MDA-MB-468 breast cancer by slowing down their progression when treated with pomegranate seed oil. 

Because of all the above-mentioned significant activities, pomegranate peels can be added to commercial tomato and orange juices that contain strawberries. The extract of dried pomegranate peel substantially boosted the antioxidant content of both flavored juices. Due to the astringent flavor of pomegranate peel, the maximum extract concentration in the juices had the lowest sensory acceptability. According to the results, adding 0.5% dried pomegranate peel extract to strawberries was approved in tomato and orange juice [21]. This study investigated the antioxidant, total phenolic, tannin, and phytic acid levels of four fruit peels (pomegranates, grapes, apples, and mocha palms). The maximum antioxidant activity and phenolic content were found in pomegranate peel. It was somewhat poor in flavonoids and vitamin C. The sensory assessment of guava juice supplemented with pomegranate peel did not vary from the control. Incorporating pomegranate peel into guava juice is therefore highly well received, increases the antioxidant components, protects health, and enhances quality of life [22]. Altunkaya et al. [23] formulated apple juice supplemented with pomegranate peel extract (PPE) at concentrations ranging from 0.5% to 2%. The most pomegranate peel was found in the juice, as well as the most antioxidant activity.

In comparison, only 0.5 g PPE per 100 mL was determined to be acceptable for color, smell, and taste. The conclusion was that PPE-enriched apple juice is superior to regular apple juice. Therefore, adding PPE at concentrations between 0.5% and 1.0% proved acceptable and generated a healthier product without altering apple juice’s sensory qualities or toxicological safety.

No available studies cast light on the pomegranate pomace and its application in the food industry, in which it may be more valuable than peels. Therefore, this study aimed to complete the evaluation of pomegranate pomace concerning the phenolic content by LC/MS, and antioxidant, anticancer, antibacterial, antifungal, and antiviral properties. Additionally, it encourages using natural and safe alternatives to control pathogenic microorganisms and component oxidation. Therefore, highly functional pomegranate pomace extract was added to the strawberry-yogurt smoothie to enhance its color, flavor, and texture, besides extending its shelf life and quality.

## 2. Materials and Methods

### 2.1. Preparation of Pomegranate Pomace Aqueous Extract (PP)

Pomegranate pomace was obtained from the juice company. The PP was homogenized and washed with tap water, followed by distilled water. The pomace was dried in an oven at 50 °C for 2 days, then powdered and kept dry until extraction. Ten grams of pomegranate pomace powder was stirred in 150 mL of sterilized distilled water for 12 h, then filtered by Whatman No. 1 paper. The filtered extracts were centrifuged at 4000 rpm/15 min and kept at 4 °C.

### 2.2. Proximate Analysis of Pomegranate Pomace Powder

Moisture, protein, fat, and ash levels were determined according to AOAC guidelines [24]. The carbohydrates were determined by differential analysis. As per Saad et al. [25], the water-absorption capacity (WAC) and oil-absorption capacity (OAC) were evaluated as follows: PP powder (1 g) was stirred in 10 mL of water or oil for 30 min, then centrifuged at 7500 rpm/30 min in weighted test tubes. The residue was weighed, and WAC and OAC were estimated as mL of water or oil retained per gram of sediment.

### 2.3. Phenolic Profile in PP Extract

#### 2.3.1. Total Phenols and Total Flavonoids Content

Total phenolics were determined in PP extracts by the Folin–Ciocalteu method described by Saad et al. [26], and the data were calculated as mg of gallic acid equivalents (GAE)/g. The total flavonoid content was estimated using the AlCl_3_ colorimetric method following Namir et al. [27]. The data were estimated as mg quercetin equivalents (QE)/g.

#### 2.3.2. High-Resolution LC/MS Profile of Phenolic Content of PP Extract

Accela U-HPLC system (Thermo Fisher Scientific, USA) was used to identify the phenolic compounds in PP. LC/MS data were obtained by an Executive Orbitrap mass spectrometer linked with the HPLC system. The compounds were separated following Colantuono et al. [28]. Separation column (Gemini C18-110, Phenomenex, Torrance, CA, USA), 150 × 2.0 mm, 5 µm and mobile phases were (A) aqueous Formic acid 0.1% and (B) Formic acid in acetonitrile 0.1%. The column was heated to 30 °C, and the flow rate was adjusted at 0.2 mL/min. The PP extract was dissolved in a mixture of methanol and water (50:50). Then, 10 µL of the prepared sample was injected into the column. Negative ionization methods collected MS data in the mass range of 100–1400 m/z. The capillary temperature was 275 °C, the auxiliary gas had a pressure of 15 arbitrary units, and the sheath gas had a pressure of 30 arbitrary units. Linear calibration curves of PC (1–50 µg/mL) and EA and GA (0.1–5 µg/mL) were established. With a mass tolerance of ±5 ppm, the metabolites were identified using accurate mass values to the fifth decimal place.

### 2.4. Pomegranate Pomace Extract Activity

#### 2.4.1. DPPH-Radical-Scavenging Activity

The antioxidant activity was determined using the DPPH technique, as described by Abdel-Moniem et al. [29]. A total of 0.5 mL of ethanolic DPPH was added to 1 mL PP extract and incubated for 30 min in the dark; the produced color was read spectrophotometrically at a wavelength of 517 nm. The absorbance was applied as in Equation (1). The lowest concentration that scavenges 50% of DPPH radical was calculated as IC_50_ [30].
(1)$$\%\;DPPH\;scavenging\;activity=\frac{Abs\;control-Abs\;sample}{Abs\;control}\times 100$$

#### 2.4.2. Anticancer

The lung carcinoma cell line (A-549) was assayed for cytotoxicity; the doxorubicin reference standard was used as a positive control. According to Mosmann [31] and Gomha et al. [32], the cytotoxicity evaluation was carried out using a viability assay. The apoptosis-inducing potential of PP extract was assessed by measuring its caspase-3 activity. After treating the 25 cm^2^ flasks with DMSO or PP at the right doses and times, the A-549 cells were collected. For each experiment, 2 × 10^6^ cell lysates were used and analyzed as specified. After 45 min of incubation with substrate Z-DEVD-R110 in a microplate reader, fluorescence was seen, and the excitation and emission were measured at 485/530 nm [33].

#### 2.4.3. Antimicrobial

Twelve pathogenic microorganisms were employed for the antimicrobial screening test. *S. aureus*, *L. monocytogenes*, *B. cereus*, *P. aeruginosa*, *K. pneumonia*, *E. coli*, *A. niger*, *A. flavus*, *C. glabrata*, *C. albicans*, *C. davenportii*, and *P. expansum*. These strains were selected based on the microbial count of spoiled yogurt samples. It was found during microbial examination with a light microscope and biochemical and morphological definitions that these isolates are the most isolates that cause smoothie spoilage. These isolates were confirmed by identification at the gene level through isolating DNA and using PCR to detect genes. The bacterial isolates were identified based on 16S rRNA and the fungal isolates on 18S rRNA gene sequence analysis. Sequencing was performed via the automated DNA sequencer (ABI Prism 3130 Genetic Analyzer by Applied Biosystems Hitachi, Japan). Genomic DNA was obtained by the hexadecyltrimethylammonium bromide (CTAB) technique, and the integrity and level of purified DNA were established by agarose gel electrophoresis. The DNA level was customized to 20 ng/µL for PCR amplification. The forward primer used with the isolates is (5 AGA GTT TGA TCC TGG CTC AG 3), and the reverse is (5 GGT TAC CTT GTT ACG ACT T 3). PCR products were isolated by electrophoresis on 1.5% agarose gels stained with ethidium bromide and documented in the Alphaimager TM1200 documentation and analysis system. The obtained polymorphic differences were analyzed via the program NTSYS-PC2 by assessing the distance of isolates by Jaccard’s Similarity Coefficient. All the bacterial strains were maintained by sub-culturing on nutrient agar or potato dextrose slant and stored at 4 °C. The PP extract was evaluated for its antibacterial and antifungal activity by the well-disc-diffusion method, according to Dahham et al. [34] and El-Saadony et al. [35]. Next, 50 mL of molten Mueller–Hinton agar (MHA) or potato dextrose agar (PDA) was poured into plates. A loopful of bacterial or fungal inoculum was spread over the surface of the plates. Then, 8 mm wells were punched into each plate and filled with 50 μL of PP extract concentrations (200, 400, 600, 800, and 100 µg/mL), and negative control wells were filled with water. MHA and PDA plates were incubated at 37 °C for 24–48 h (bacteria) or 28 °C for 5 days (fungi) [36]. The resultant inhibition zone diameters (mm) indicated the antimicrobial activity [37]. The MIC, MBC, and MFC values were determined by Saad et al. [25].

#### 2.4.4. Antiviral

ACE2 was created as a recombinant protein in human cells. SARS-CoV-2 Inhibitor Screening Assay kit (Adipogen, San Diego, CA, USA) was used to inhibit the binding between the Spike and ACE2. The SARS-CoV-2 antiviral activity of the PP extract was evaluated as follows: 100 µL of Spike RBD was added to each well of a 96-well plate and kept for 16 h at 4 °C. Then, 100 µL of PP concentrations (200, 400, 600, 800, and 1000 µg/mL) was added to wells containing Spike and incubated for one hour at 37 °C in the presence of Inhibitor Mix Solution ((0.2% BSA, 0.05% Tween R 20 in PBS, and biotin-conjugated ACE2 (0.5 g/mL)). After incubation, HRP-labeled streptavidin (1:200 dilution) was added to each well and incubated for one hour at room temperature. To end the reaction, 100 µL of tetramethyl benzidine was added and incubated for 5 min. The absorbance was read at 450 nm by the microplate reader (BioTek Elx808, Winooski, VT, USA).

### 2.5. PP-Strawberry-Yogurt Smoothie Processing and Preservation

#### 2.5.1. Preparation of Strawberry Smoothie

Fresh strawberry fruits were washed, cleaned, and processed for juice production using a Moulinex mixer following Saad et al. [26] with some modifications. In a blender, 50 mL of PP extract (25 g/100 mL) at different concentrations and yogurt (40 g) were mixed well with filtered strawberry juice (100 mL). The water and sugar were added as shown in Table 1. The smoothies, SYSC, PP-SYS1, PP-SYS2, PP-SYS3, and PP-SYS4, were transferred into 250 mL bottles and pasteurized in a TOMY Sx-700 autoclave (Tokyo, Japan) at 95 °C for 2 min under 50 MPa. The bottles were cooled and stored at 4 °C for two months.

#### 2.5.2. Physiochemical Properties

The pH values of the smoothies, SYSC, PP-SYS1, PP-SYS2, PP-SYS3, and PP-SYS4, were calculated using a pH meter (pH 211, HANNA, Nusfalau, Romania). Standard method 942.15 was used to estimate the titratable acidity, which was recorded as a percentage of citric acid. A refractometer was used to calculate the TSS. AOAC [24] was also used to assess vitamin C concentration. The Gerber technique was used to determine fat content; an increase in fat percentage from 0.4 to 4% while keeping protein constant improved texture, stability, and perceived viscosity. The total sugars were estimated by El-Saadony et al. [38].

#### 2.5.3. Color Differences and Sensory Properties

Hunter spectrophotometer (AA Color Flex EZ) was used to determine the color parameters (*L**, *a**, and *b**) of PP-SYS (HunterLab, Murnau, Germany). The following parameters were considered: Equation (2) calculated the color difference (∆E) between smoothie samples during the storage period [39].
(2)$$\Delta E=\sqrt{{\left(\Delta L\right)}^{2}+{\left(\Delta a\right)}^{2}+{\left(\Delta b\right)}^{2}}$$

The sensory characteristics (color, flavor, texture, and overall acceptability) of PP-SYS were estimated using a 9-point hedonic scale. PP-SYS samples were randomly coded and served in one-use cups to 30 semi-trained panelists, 20 males and 10 females (aged 21–30), in a fluorescent-lit laboratory with an air-conditioning temperature of 23 °C. Water was used to change the mouth contact areas after each judgment to avoid interfering with the results.

#### 2.5.4. Microbial Load

Ten PP-SYS samples were stirred in 90 mL of peptone water for 30 min to obtain a 10^−1^ suspension. Serial dilutions were generated up to 10^−8^. The dilutions were put into disposable Petri plates with a specific medium [40]. The total viable count (TBC) was determined after 24 h incubation at 30 °C using plate count agar. The results for microorganisms were converted to logarithms (CFU/mL).

### 2.6. Statistical Analysis

A one-way ANOVA test was used to distinguish the significant differences between sample means at a probability level (*p* ≤ 0.05). The LSD test determined significant differences between means. All statistical calculations were performed with SPSS 20 for Windows.

## 3. Results and Discussion

### 3.1. Chemical Composition

The findings revealed that pomegranate pomace is a rich source of carbohydrates and fiber with very low-fat content. Table 2 shows the proximate analysis of pomegranate pomace, which comprised 0.61% fat, 4.12% ash, 6.98% moisture, 9.11% protein, and 20.66% fiber, with total carbohydrates accounting for 58.52% of the sample. There are no studies on pomegranate pomace, but the findings are comparable with those of Ranjitha et al. [41], except for pomegranate peels that contain 0.85% fat, 4.32% ash, 7.27% moisture, 3.74% protein, 17.31% fiber, and 66.51% carbohydrates. Moreover, El-Hadary and Taha [42] found that the chemical composition of pomegranate peel was crude carbohydrates (78%), fiber (12%), protein (3.5%), ash (3.4%), and lipids (2.25%). 

The high content of carbohydrates and fibers in PP influences the WAC and OAC values; in this study, the WAC of pp was 8.21 mL/g, compared to lower WACs in cantaloupe waste, dates, pear, and tomato pomace with values of 6.17, 5.70, 4.90, and 4.12 mL/g, respectively [43]. The OH in polysaccharide chains can build hydrogen bonds with water, enhancing fiber-rich materials’ capacity to retain moisture [44]. Regarding the OAC of PP, it was 7.77 mL/g, which is higher than lemon byproducts (6.60 mL/g) and tiger nut pomace (6.90 mL/g) [45]. The OAC is crucial since lipids serve as taste preservers and enhance the texture of food. The OAC is also a technological property related to the chemical structure of the plant polysaccharides. It depends on surface properties, overall charge density, thickness, and hydrophobic nature of the fiber particle [46]. Regarding the color parameters of PP, *L** and *b** values are high, which is a good indicator for enhancing the color when applied to food formulation.

### 3.2. Phenolic Compounds in Pomegranate Pomace Aqueous Extract (PP)

Phenolics are chemicals generated from the secondary metabolism of plants that are frequently used for their biological effects, notably their antioxidant properties. Phenolic compounds are antioxidants with biological and chemical actions, the most significant of which is free radical scavenging [47]. Figure 1 shows the significant (*p* ≤ 0.05) phenolic content (total phenolic content and total flavonoids) in PP. The concentration of TPC and TF significantly (*p* ≤ 0.05) increased in a concentration-dependent manner when PP (1 mg/mL) contained 205 mg of gallic acid equivalent/mL and 110 mg of quercetin equivalent/mL. There are no studies on the antioxidant content of PP extract; previous studies have focused on pomegranate peels and waste extracts with a high concentration of phenolics and flavonoids; hence, pomegranate peels can be considered a major source of antioxidants. The results may agree with Yassin et al. [48], who found that acetone extract of pomegranate peels contained 246 and 227.7 mg GAE/g. In comparison, the methanolic extract of pomegranate peels has a maximum total phenolic content of 18.89 g/100 g and a total flavonoid content of 13.95 mg QE/kg [42]. There is a preference for our PP in phenolic content, with a relative increase of 11%.

To determine the exact and active phenolic compounds in pomegranate pomace, high-resolution LC/MS was processed (Table 3). The detected compounds belong to three groups: ellagitannins (ETs), ellagic acid derivatives (EA), and gallic acid (GA); their values are 479.35, 24.29, and 1.20 g/L, respectively. The main compound in LC/MS profile is punicalagin (199 g/L), followed by granatin B (60 g/L) and pedunculagin A (52 g/L). In the LC/MS profile, causarinin, punicalin, galloyl-HHDP-hexoside, granatin B, pedunculagin A, and pedunculagin B were expressed as punicalagin (PC) equivalents. EA equivalents were EA hexoside, pentoside, and deoxyhexoside. Gallic acid was calculated with the corresponding standard. Total polyphenols accounted for 500 g/L; PC derivatives represent 95% of total phenolics, while EA derivatives and GA represent only 5%. Similar results were observed by Zivkovic et al. [49]; they found that pomegranate wastewater extract contains punicalin (197.13 mg/g), punicalagin (54.23 mg/g), and ellagic acid (25.42 mg/g) found to be dominant in peel extract. On the other hand, Coronado-Reyes et al. [50] found that pomegranate peel is a more important source of punicalagin and ellagic acid than aryls, where the punicalagin content of the crust ranged from 114.6 to 282 mg/g dry weight. The ellagic acid content varied from 1.07 to 2.49 mg/g dry weight. Additionally, a previous study on pomegranate aril showed that ellagic acid was detected in different extracts, and it was the highest detected phenolic compound with 34.5 (μg/mg) when a combination of ethanol: ether: water (8:1:1) was used [51]. These compounds can potentially be used in foods and medical applications [52].

### 3.3. Activity of Pomegranate Pomace Aqueous Extract

#### 3.3.1. Antioxidant

The antioxidant potential of the pomegranate pomace aqueous extract was evaluated according to a 1,1-diphenyl-2-picrylhydrazyl (DPPH) assay; the SC_50_ (antioxidants needed to reduce 50% of the initial DPPH concentration) was identified. Effective free radical-scavenging activity was recorded in a concentration-dependent manner, with considerable SC_50_ = 200 µg/mL (Figure 2). Increase in the extract concentration resulted in increased inhibition percentage (decrease in the concentration of DPPH) and reached almost complete inhibition (95%) at 1 mg/mL of PP compared to ascorbic acid. Lower DPPH radical-scavenging activity values were found with 93% in PP ethanolic extract [42], 83.6% in PP ethyl acetate extract [53], and 78.23% in pomegranate waste methanolic extracts [54]. Bopitiya and Madhujith [55] concluded that significant amounts of phenolic and flavonoid compounds (such as ellagic and gallic acids) were attributed to the antioxidant properties of pomegranate peel extracts.

Considerable antioxidant activity is essential to inhibit lipid oxidation in the food system. Free radicals are unstable and can react with biomolecules of living cells, causing various diseases such as cardiovascular diseases and cancer [56]; the antioxidants help the body to protect itself against this oxidative damage. The considerable amounts of hydrolyzable tannins in pomegranate peel are responsible for their antioxidant power [57]. Mekni et al. [17] found a significant correlation between the best antioxidant activity of various pomegranate extracts and the extract’s highest phenolic and flavonoid contents; furthermore, this antioxidant activity can be related to the presence of different functional groups, such as hydroxyl and carbonyl groups.

#### 3.3.2. Anticancer

In vivo anti-proliferative activity was assessed against lung carcinoma (A-549) using different concentrations of PP extract. The results indicated that the extract displayed potent anticancer activity and cytotoxic effects against A-549 lung carcinoma. The doxorubicin drug was also tested as a positive control for comparison. As illustrated in Figure 3A,B, increasing inhibition of cell viability with increasing extract concentrations was observed (dose-dependent manner). Maximum growth inhibition of 86% was detected for lung carcinoma cells at a dose of 1000 μg/mL. The 50% inhibition of cell viability (IC50) for A-549 was 250 μg/mL, representing a valuable result compared to the IC_50_ value of doxorubicin of 300 μg/mL. Compared to carcinoma cells, a less cytotoxic effect was recorded against normal lung cells. The anticancer activity of the pomegranate peel ethanolic extract was carried out against the oral cancer cell line (KB 3-1), and this peel extract exhibited promising anticancer properties. The MTT assay showed 94.53% inhibition on the oral cancer cell line, and the clonogenic assay showed a decrease in the colonies after treatment with the peel extract [58]. Moreover, against various kinds of cancer, for example, in prostate cancer and renal cell carcinoma tissues grown in vitro, pomegranate extracts inhibit the activity of NF-kB [59]. Punicalagin, generated from pomegranate extracts, has anti-proliferative activity by inducing apoptosis of cancer cells such as lung carcinoma and breast and cervical cancer cell lines [60]. Induction of the apoptosis process by punicalagin through caspase activation and poly ADP ribose polymerase (PARP) inhibition. Caspase-3, -8, and -9 protein expressions exhibited a dose-dependent increase after 50 and 75 µM punicalagin treatment. Furthermore, A549 cells treated with 50 and 75 µM concentrations of punicalagin showed cleavage of PARP protein [61].

Figure 3C shows that the activity of caspase-3 increased with cancer cell death, showing high activity when lung cancer cell lines were treated with PP extract (1 mg/mL). Caspase-3 is essential for normal brain development and other apoptotic conditions in a manner that is tissue-, cell-, or death-stimulus-specific. It is also necessary for apoptotic chromatin condensation and DNA fragmentation in every examined cell type. Caspase-3 is therefore required for cellular disintegration and the formation of apoptotic bodies. However, it may also function before or during cell death [62]. Pomegranate fruit and its extracts have been found to combat cancer in many in vitro and in vivo studies; modulating various signaling pathways is the mechanism of action against cancer growth. The introduction of targeted medicines has revolutionized the treatment of patients with A-549 carcinoma. When combined with natural extracts with stratagems, repurposing non-cancer drugs into new therapeutic niches presents a cost-effective and efficient technique with enhanced outcomes for discovering novel pharmacological activity to potentially increase cure rates for lung and liver cancers [63,64].

#### 3.3.3. Antimicrobial

Table 4 shows the antibacterial and antifungal activities of PP extract against some foodborne and multidrug-resistant pathogens. The extract has a wide spectrum of antimicrobial activity; the inhibition zone diameters (IZDs) significantly increased *p* ≤ 0.05 in a concentration-dependent manner and were 9–35 mm against the tested bacteria compared to penicillin. The most sensitive bacteria to PP extract (1 mg/mL) were *S. aureus* with IZD, 35 mm. However, *K. pneumonia* were the most resistant bacteria to PP extract (1 mg/mL), where IZD was 27 mm, with a relative decrease of about 30% of the *S. aureus* inhibition zone.

The resistant Gram-negative bacteria have lower IDZs than the Gram-positive ones because of the unique structure of the Gram-negative bacteria’s membrane.

On the other hand, as shown in Table 4, PP extract exhibited considerable antifungal activity against *Candida, Aspergillus,* and *Penicillium* species in this study, with IZDs ranging from 13 to 35 mm compared to clotrimazole (1000 µg/mL). *A. niger* and *C. glabrata* were the most resistant fungi to PP extract, while *A. flavus* and *P. expansum* were the most vulnerable. Appendix A shows the IZDs image of PP extract 0.6, 0.8, and 1.0 mg/mL against tested bacteria (*S. aureus*, *L. monocytogenes*, *B. cereus*, *P. aeruginosa*, *K. pneumonia*, *E. coli*), Candida (*C. glabrata*, *C. albicans*, *C. davenportii*), and fungi (*A. niger*, *A. flavus*, *P. expansum*). Kesur et al. [12] demonstrated that the high efficiency of the methanolic extract of pomegranate peel against the *Bacillus cereus, Bacillus megaterium, Bacillus subtilis, Escherichia coli*, *and Aspergillus niger* showed inhibition zones between 6 and 17 mm. At the same time, no activity was detected by chloroform extract against the same microorganisms. On the other hand, pomegranate waste extract has powerful antifungal activity against *Candida gelberta* and *Candida apis* [65].

The PP extract successfully inhibited the bacterial and fungal isolates in the 45–160 µg/mL concentration range while killing the tested isolates with 80–290 µg/mL concentrations (Table 5). Kesur et al. [12] found that the minimum inhibitory concentration of the methanolic extracts of peels of pomegranate on *B. cereus*, *B. megaterium*, *P. vulgaris*, and *P. aeruginosa* was observed in the range of 50,0000 to 12,500 (µg/mL). In contrast, the MIC of peel extracts on *B. subtilis*, *S. typhi*, *S. typhi A*, and *S. typhi B* was found to be in the range of 50,000 to 25,000 (µg/mL), while MIC against fungi is higher than that. An efficient antimicrobial effect expressed as MIC of the crude extract was observed against the studied strain, for which we recorded 312.5 μg/mL. It was reported by Oliveira et al. [66] that strong inhibitors have MIC values below 500 µg/mL, moderate inhibitors have values between 600 and 1500 µg/mL, and weak inhibitors are above 1600 µg/mL. Hasan et al. [67] confirmed that pomegranate flavonoids and phenolic compounds are extracted more efficiently with methanol due to their polarity. Naziri et al. [68] showed that active antimicrobial substances have different solubility in various solvents and that using different solvents for extraction changes the extent of the antibacterial effect of the extract.

Pomegranates are a good source of potentially bioactive phytochemicals such as phenolics, flavonoids, anthocyanins, alkaloids, and tannins [8]. The results of this study clearly showed a good relationship between the IDZs of the studied bacteria and pomegranate varieties. Rosas-Burgos et al. [69] consistently observed different inhibition levels for the studied pomegranate peel extracts, where the *Salmonella* strain was the most sensitive. They concluded that the highest punicalagin and ellagic acid concentrations were detected in sour and sweet pomegranate cultivars with the highest inhibitory activity. Vasconcelos et al. [70] numerous microbial enzymes in culture filtrates or pure forms are inhibited by tannic, and the toxicity of tannins may be connected to their effect on microbes’ membranes. High antibacterial activity for the studied methanolic extracts was observed, while chloroform extracts were less effective, with highly significant differences. Suresh et al. [71] observed a gradual decline in the protein content of bacterial cells incubated with different concentrations of pomegranate peel extract (PPE) compared to the control.

Based on the polynomial model in (Figure 4), employing a higher concentration of PP, i.e., 2 mg/mL, will increase the IZDs by 58.6, 96.4, 20.3, and 42.5% against *Listeria monocytogenes*, *E. coli*, *C. albicans*, and *A. flavus*, respectively, compared to PP (1 mg/mL). We noticed that *C. albicans* were more sensitive to PP extract (2 mg/mL), followed by *E. coli.*

#### 3.3.4. Antiviral

Figure 5 shows the inhibition activity of PP extract on the binding between Spike and ACE2 compared to a SARS-CoV-2 inhibitor test kit. At doses of 200, 400, 600, 800, and 1000 µg/mL, PP extract inhibited the interaction between Spike and ACE2 by as much as 75.6%. This impact was dose-dependent compared to the positive control, AC384, a monoclonal antibody that prevented the binding between Spike and ACE2 by identifying ACE2 itself and inhibiting 70% of the interaction. Surucic et al. [72] found that punicalagin in pomegranate peel extract inhibited 50% of S-glycoprotein and ACE2 contact only in the highest concentration sample (1 mg/mL).

There are no studies on the antiviral activity of PP extract; however, most earlier investigations focused on pomegranate peel extracts, which showed antibacterial and antiviral activity and suppressed influenza and herpes viruses, which can also be employed effectively as an antiviral agent against SARS-CoV-2. In vitro, the aqueous–alcoholic pomegranate peel extract stopped SARS-CoV-2 S-glycoprotein from binding to ACE2, showing that the extract can stop SARS-CoV-2 from entering the host cells [73].

### 3.4. Changes in Physiochemical Properties of PP-Strawberry-Yogurt Smoothie

#### 3.4.1. Physiochemical Properties

The functional milk beverage fortified with pomegranate pomace extract during storage influences the pH and beverage acidity. During two months of storage at 4 °C, the pH values of PP-smoothie samples decreased by 8–13% compared to the control smoothie; therefore, the pp-smoothie acidity increased by 16–21% compared to the control. Thus, in agreement with Al-Hindi and Abd El Ghani [74], pH levels of the functional milk beverages with and without POPE decreased significantly from 4.5 to 4.1 with increasing storage time. The higher counts of viable bacteria at the end of the storage period resulted in lower pH levels in the functional milk beverages fortified with or without POPE. In addition, the beverage’s acidity increased markedly from 0.2 to 0.35 with increasing storage time.

Massive decreases in vitamin C content in the control sample (133%) after 2 months of storage, but these reductions were considerably inhibited (*p* ≤ 0.05) in PP-smoothies by 11–55% over control. The high phenolic content in PP extract inhibited the oxidation of vitamin C, and total sugars were decreased with storage period by up to 53% in the control sample, but these reductions decreased in PP-SYS1, PP-SYS2, PP-SYS3, and PP-SYS4 by 4–28% compared to the control. The high phenolic content in PP extract inhibited the fermentative action of microbes against sugars. The reductions in sugars were followed by an increase in TSS (28–38%) in PP smoothies compared to the control. In addition, the fat content was elevated based on pomegranate pomace concentrations (Table 6). During three weeks of cold storage, a substantial increment in fat content was detected between untreated and treated yogurt (*p* < 0.05) [3]. Saad et al. [75] observed the same trend in cucumber beverages supplemented with polyphenolic extracts, i.e., decrements in vitamin C and total sugars while increasing TSS percentage. They noticed that the decrements in vitamin C and total sugars were mitigated by polyphenolic extract addition.

Strawberry-yogurt smoothie (SYSC) supplemented with pomegranate pomace extracts 0.4 mg/mL (PP-SYS1), smoothie supplemented with pomegranate pomace extract 0.8 mg/mL (PP-SYS2), smoothie supplemented with pomegranate pomace extract 1.2 mg/mL (PP-SYS3), smoothie supplemented with pomegranate pomace extract 1.6 mg/mL (PP-SYS4).

#### 3.4.2. Total Phenolic Content

Polyphenols have a crucial role in the color and flavor of foodstuffs while being very volatile and easily oxidized [76]. Adding PP extract with different levels (0.4, 0.8, 1.2, and 1.6 mg/mL) to the tested smoothie enhanced the antioxidant activity from 50% in the control smoothie to 88% in PP-SYS4 (data not shown). The antioxidant activity of smoothies was enhanced due to the presence of phenolics and flavonoids that were boosted by the PP extract. The total phenolic and flavonoid content in PP-SYS4 increased by 170% compared to the control at zero months of storage. However, after 2 months of storage, the activity further increased, estimated at 98%, despite decreasing the phenolic and flavonoid content by 14% (Figure 6). No studies have focused on using PP extract as an additive in foods. However, our findings may be correlated with Ahmed et al. [77], who added pomegranate peels to yogurt and observed their effects on total phenolic contents and antioxidant activity of the product; they found an average TPC in yogurt of (4.07 ± 0.37) mg GAE/g. However, the maximum content was detected in pomegranate yogurt (4.64 mg GAE/g compared to the control, which was 3.39 mg GAE/g). Antioxidant activity examination reveals that an average (70.58%) was calculated. Meanwhile, extreme activity was diagnosed in the pomegranate yogurt sample (83.87%), having 9% freeze-dried PPP.

Due to the abundance of phenolic components in several plant extracts, developing natural alternatives to synthetic preservatives in food is of special interest because these compounds possess antioxidant and antimicrobial properties that inhibit the process of lipid peroxidation in fatty foods and scavenging free radicals and prevent the growth of microorganisms [75,78,79].

#### 3.4.3. Color Difference and Sensorial Properties

Analysis of the nutritional values and the quality of ingredients of the beverages are very important to ensure the safety and quality of the beverages. Sensory investigation using color, taste, odor, and textural analysis efficiently assesses the final beverage quality [80]. Based on the color analysis of PP in Table 2, the extract has a high redness value that enhances the whiteness and red color of the strawberry-PP yogurt smoothie. Nonetheless, at the end of storage, the color parameters faded (data not shown), but the addition of PP extract mitigates the color reduction, and that is clear in Figure 7, where color change (ΔE) value was observed as limiting the color change in PP-supplemented smoothies; the ΔE value in the control was 6.14; however, in PP-SYS4, it decreased by 22%. Trigo et al. [81] found that with the addition of pomegranate peel extract to carrot beverage, the *whiteness* was not affected by PPE, but heat treatment led to higher *L** levels compared to the HPP treatment. The increase in *L** values was noticed in the first week of storage, but the *L** remained constant within the next few days. Concerning ΔE, this difference was evident throughout the entire storage period. For example, on day 7, the Δ*E** changed between 1.2 and 2.0 for HPP-treated juice and between 22.1 and 32.1 for TP-treated juice.

Table 7 shows the sensory properties of tested smoothies. The results indicated that PP-SYS4 gained the highest scores by panelists, i.e., flavor and color (9), texture (8.8), and over acceptability (8.9) after a month of storage at 4 °C, but a slight decrease was noticed and reflected on for acceptability values that declined by 3% compared to the control, which decreased by 9%. The results for the other smoothies are in between PP-SYS4 and the control. Ahmed et al. [77] noticed low sensorial scores in relation to the yogurt with pomegranate peels, which were reduced by 9%, but decreasing the addition level increases the sensorial scores. Sensory evaluation of food products is important in determining what consumers might like. The main way to judge a juice’s quality is by color [82].

### 3.5. Microbial Load Changes during Storage of Smoothie

The microbial load in all smoothie samples was significantly increased with the storage period; however, it decreased with the gradual addition of levels of PP extract. The microbial load in control samples was 3.5 CFU/mL and increased by 56% after two months. However, in PP-SYS4, the microbial load significantly decreased (*p* ≤ 0.05) by 37% compared to the control after 2 months of cold storage (Figure 8). Ebrahimain et al. [83] studied microbiological stability and sensory properties of traditional Iranian butter with pomegranate peel extract incorporated into it, and they found that PPE could reduce the bacterial count in butter from 9.25 log CFU to 6.30 log CFU after 90 days of storage, while with increasing storage time, the total number of bacteria was increased in the treatments; this increase is most evident when dairy products are made with poor safety quality.

Many studies have indicated that phenolic compounds in PP extract contribute to the direct inhibition of bacterial pathogens and higher antioxidant activity [84]. The ultracellular damage of tested bacteria treated with sour pomegranate peel extract was examined using TEM, illustrating severe damage in the bacterial cells and confirming the extract’s potential antibacterial and lethal activity. Karaman et al. [85] also showed that lipophilic parts in phenolic acids permeate through the cell membrane by passive diffusion, enhancing membrane permeability by reducing intracellular pH. The triggering of protein denaturation, explained by Li et al. [14], confirmed that damage to the bacterial cell membrane is one of the possible modes of action for the phytochemical elements, followed by releasing intracellular material and cell death.

## 4. Conclusions

There is a current need to renew attention in looking for natural resources of bioactive active compounds to be used as safe food additives. Isolation of such compounds from agro-industrial wastes will provide a cost-effective solution for controlling the spread of human pathogens and improve waste reduction. Given that, pomegranate pomace was found to exert a marked reduction in multidrug resistance and food pathogens and possesses great potential to scavenge DPPH free radicals with high antioxidant capacity, besides its potent antitumor activity in lung cell lines and potential cure of COVID-19. Adding PP extract (1.2 or 1.6 mg/mL) to SYS samples enhances color, texture, and quality and introduces a functional beverage that may benefit many patients. Considering all these details, we believe that pomace of *Punica granatum* may provide a safe and efficient alternative-perspective useful additive in functional food. We hope to continue further explorations of this attractive botanical species.

## Figures and Tables

**Figure 1 bioengineering-09-00735-f001:**
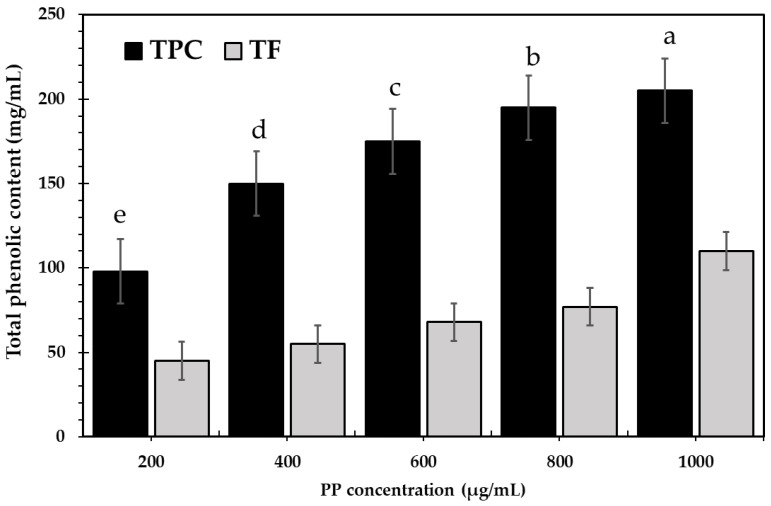
Total phenolic and flavonoid content in pomegranate pomace aqueous extract (PPE). TPC, total phenolic content; TF, total flavonoids; PP, pomegranate pomace. Data are presented as mean ± SE. Lowercase letters in the same columns indicate significant differences *p* ≤ 0.05.

**Figure 2 bioengineering-09-00735-f002:**
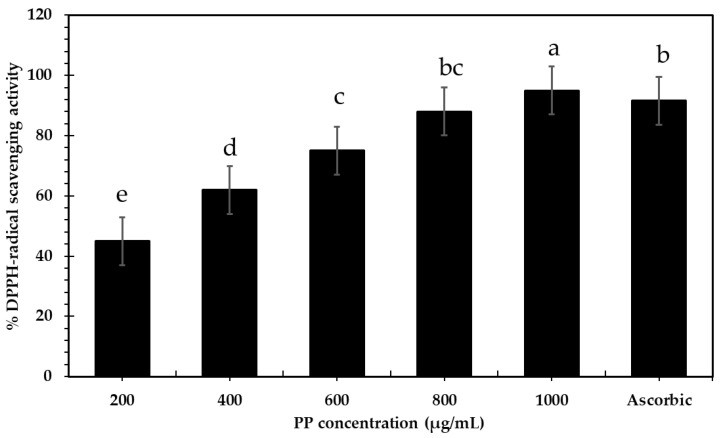
DPPH-radical-scavenging activity of pomegranate pomace aqueous extract (PP). Data are presented as mean ± SE. Lowercase letters in the same columns indicate significant differences *p* ≤ 0.05.

**Figure 3 bioengineering-09-00735-f003:**
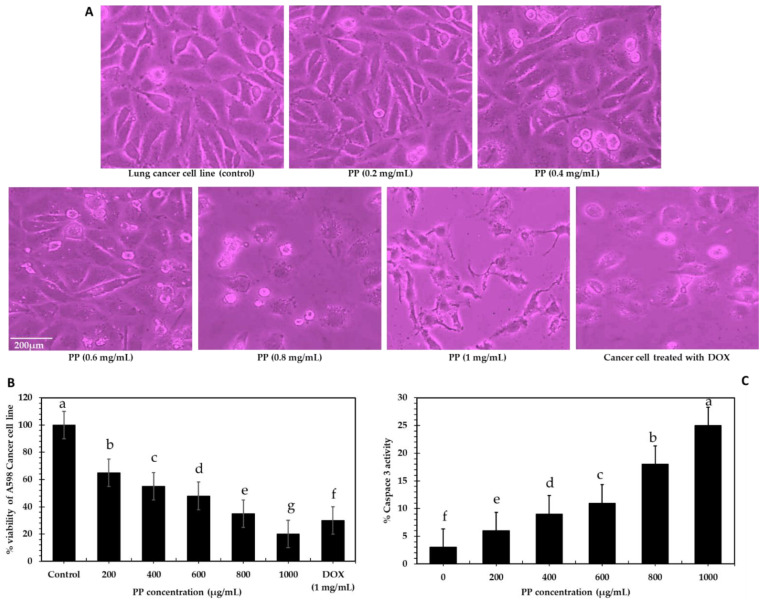
(**A**) Microscopic image of the effect of PP at different levels (0.2–1 mg/mL) on the lung cancer line viability compared to the negative and positive control, (**B**) histogram of PPE effect on cancer cell lines viability, (**C**) % caspase 3 activity in response to cancer cell death. Data are presented as mean ± SE. Lowercase letters in the same columns indicate significant differences *p* ≤ 0.05.

**Figure 4 bioengineering-09-00735-f004:**
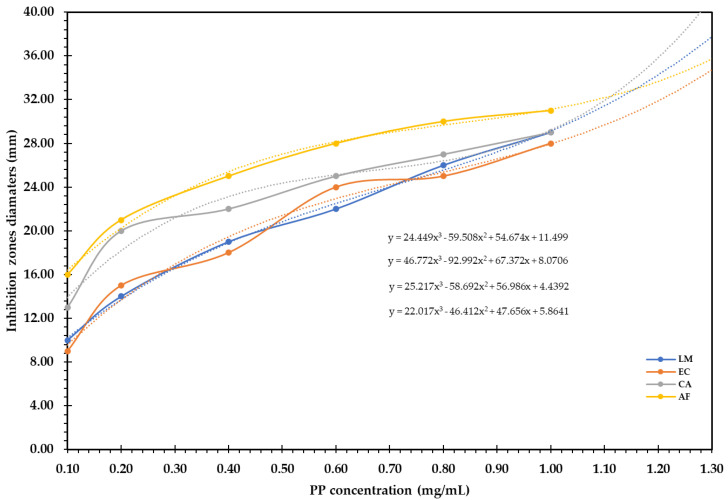
Polynomial model for forecasting the antimicrobial activity with increasing PP concentrations.

**Figure 5 bioengineering-09-00735-f005:**
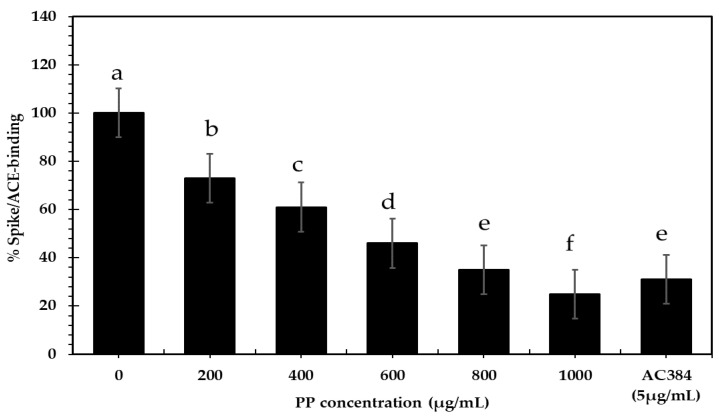
PP concentrations (200–1000 µg/mL) affect Spike/ACE2 binding compared with control and antibody inhibitor AC384 (5 µg/mL). ACE2, angiotensin-converting enzyme 2; PP, pomegranate pomace extract. Data are presented as mean ± SE. Lowercase letters in the same columns indicate significant differences *p* ≤ 0.05.

**Figure 6 bioengineering-09-00735-f006:**
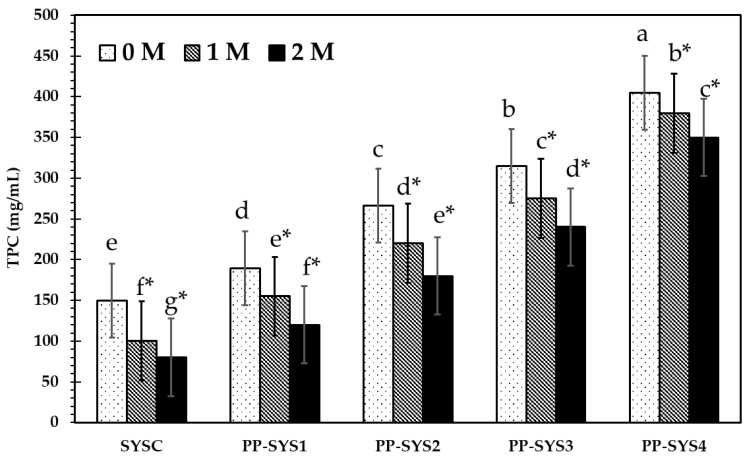
Fluctuations in total phenolic content (TPC) of smoothie supplemented with pomegranate pomace extract at different addition levels (0.4, 0.8, 1.2, 1.6 mg/mL) during two-month storage at 4 °C. Lowercase letters above columns indicate significant differences between SYS samples. Lowercase letters with a star indicate a significant change during the storage period for each SYS.

**Figure 7 bioengineering-09-00735-f007:**
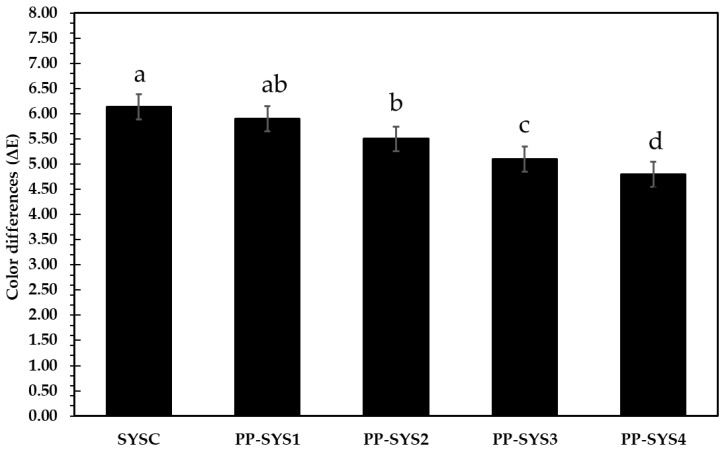
Changes in the color of the smoothie supplemented with pomegranate pomace extract at different addition levels (0.4, 0.8, 1.2, 1.6 mg/mL) during two months of storage at 4 °C. Different lowercase letters above columns indicate significant differences.

**Figure 8 bioengineering-09-00735-f008:**
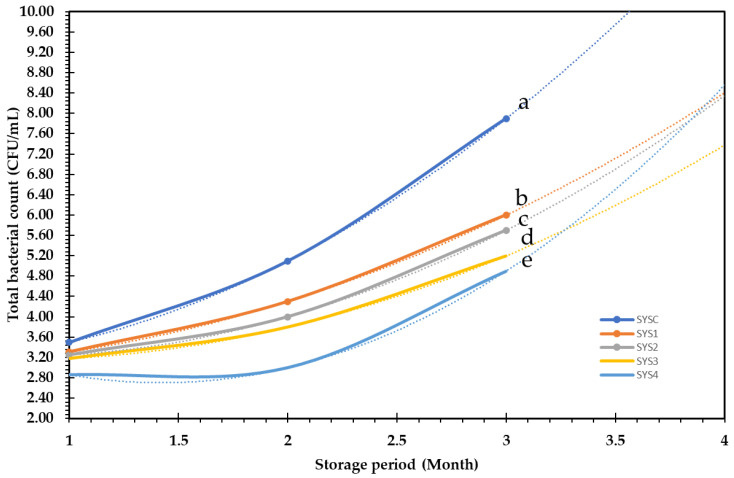
Total bacterial count (TBC) in smoothie supplemented with pomegranate pomace extract at different addition levels (0.4, 0.8, 1.2, 1.6 mg/mL) during two months of storage at 4 °C.

**Table 1 bioengineering-09-00735-t001:** The ingredients of pomegranate pomace-enriched strawberry-yogurt smoothie.

Ingredients	SYSC	PP-SYS1	PP-SYS2	PP-SYS3	PP-SYS4
Strawberry Juice (mL)	100	100	100	100	100
PP-yogurt (g)	40	40	40	40	40
Sugar (g)	10	10	10	10	10
PP (mL)	˗	50	50	50	50
Water (mL)	50	˗	˗	˗	˗

Strawberry-yogurt smoothie (SYS), control (C), smoothie supplemented with pomegranate pomace extract 0.4 mg/mL (PP-SYS1), smoothie supplemented with pomegranate pomace extract 0.8 mg/mL (PP-SYS2), smoothie supplemented with pomegranate pomace extract 1.2 mg/mL (PP-SYS3), smoothie supplemented with pomegranate pomace extract 1.6 mg/mL (PP-SYS4).

**Table 2 bioengineering-09-00735-t002:** Physiochemical parameters of pomegranate pomace powder.

PP Composition	Content (g/100 g)
Chemical	
Moisture	6.98 ± 0.2
Protein	9.11 ± 0.1
Fat	0.61 ± 0.01
Ash	4.12 ± 0.3
Fiber	20.66 ± 0.8
Soluble carbohydrates	58.52 ± 0.6
Physical	
WAC (mL/g)	8.21 ± 0.1
OAC (mL/g)	7.77 ± 0.1
*L**	61.22 ± 0.5
*a**	25.52 ± 0.8
*b**	15.66 ± 0.3

Data are presented as mean ± SD, lightness (*L**), redness (*a**), yellowness (*b**), water-absorption capacity (WAC), and oil-absorption capacity (OAC).

**Table 3 bioengineering-09-00735-t003:** Phenolic compounds profile in pomegranate pomace extract (PP) achieved by LC/MS.

RT	Compound	Equivalent	MW	PP (g/L)
6.3	Gallic acid	-	169.014	1.20 ± 0.2 ^j^
6.5	Punicalin	-	781.053	65.35 ± 0.9 ^b^
7.8	Punicalagin	-	1083.059	199.85 ± 1.2 ^a^
8.2	Pedunculagin A bis-Hexahydroxy diphenic acid (HHDP)	Punicalagin	783.069	52.98 ± 0.8 ^c^
8.2	Causarinin	Punicalagin	935.080	22.96 ± 0.8 ^f^
8.5	Galloyl-HHDP-hexoside	Punicalagin	633.073	48.15 ± 0.7 ^d^
8.7	Ellagic acid hexoside	Ellagic acid	463.052	4.50 ± 0.6 ^h^
8.8	Pedunculagin B (digalloyl- HHDP)	Punicalagin	785.084	31.66 ± 0.9 ^e^
8.8	Ellagic acid dihexoside	Ellagic acid	625.105	1.21 ± 0.1 ^ij^
9.0	Lagerstannin B	Punicalagin	949.059	1.93 ± 0.2 ^i^
9.3	Granatin B	Punicalagin	951.075	62.42 ± 0.8 ^b^
9.5	Ellagic acid pentoside	Ellagic acid	433.041	3.12 ± 0.1 ^h^
9.6	Ellagic acid deoxyhexoside	Ellagic acid	447.057	3.89 ± 0.1 ^h^
10.5	Ellagic acid	-	300.999	12.55 ± 0.3 ^g^
	Total punicalagin derivatives			479.35 A
	Total ellagic acid derivatives			24.29 B
	Gallic acid			1.20 C
	Total phenolic compounds			504.84

RT, Retention time (min); molecular weight (MW), pomegranate pomace extract (PP). Lowercase letters in the same columns indicate significant differences between detected polyphenols *p* ≤ 0.05. Uppercase letters indicate the differences (*p* ≤ 0.05) between punicalagin derivatives, ellagic acid derivatives, and gallic acid.

**Table 4 bioengineering-09-00735-t004:** Antimicrobial activity of PP extract expressed as IZDs (mm) against pathogenic microorganisms.

Pathogenic Bacteria	PP Concentration (µg/mL)						
	Inhibition Zone Diameters (mm)						
	100	200	400	600	800	1000	St * (1 mg/mL)
*S. aureus*	12 ± 0.1 ^a^	18 ± 0.3 ^a^	23 ± 0.2 ^a^	27 ± 0.5 ^a^	31 ± 0.4 ^a^	35 ± 0.2 ^a^	37 ± 0.1 ^a^
*L. monocytogenes*	10 ± 0.2 ^b^	14 ± 0.2 ^cd^	19 ± 0.1 ^c^	22 ± 0.3 ^c^	26 ± 0.2 ^c^	29 ± 0.1 ^bc^	31 ± 0.5 ^c^
*B. cereus*	11 ± 0.5 ^ab^	16 ± 0.5 ^b^	20 ± 0.2 ^b^	25 ± 0.4 ^b^	28 ± 0.5 ^b^	30 ± 0.3 ^b^	33 ± 0.3 ^b^
*P. aeruginosa*	−	14 ± 0.1 ^cd^	17 ± 0.6 ^d^	22 ± 0.5 ^c^	24 ± 0.3 ^d^	27 ± 0.7 ^cd^	29 ± 0.6 ^d^
*K. pneumonia*	−	12 ± 0.3 ^d^	16 ± 0.9 ^d^	21 ± 0.2 ^cd^	23 ± 0.3 ^e^	25 ± 0.5 ^d^	27 ± 0.6 ^e^
*E. coli*	9 ± 0.2 ^c^	15 ± 0.5 ^c^	18 ± 0.6 ^cd^	24 ± 0.1 ^bc^	25 ± 0.5 ^cd^	28 ± 0.1 ^c^	30 ± 0.2 ^cd^
Pathogenic fungi	Inhibition Zone Diameters (mm)						
*A. niger*	−	18 ± 0.6 ^c^	21 ± 0.5 ^bc^	25 ± 0.2 ^bc^	26 ± 0.3 ^c^	28 ± 0.3 ^c^	30 ± 0.3 ^bc^
*A. flavus*	16 ± 0.2 ^ab^	21 ± 0.3 ^ab^	25 ± 0.1 ^a^	28 ± 0.3 ^ab^	30 ± 0.1 ^a^	31 ± 0.4 ^ab^	34 ± 0.1 ^ab^
*P. expansum*	17 ± 0.6 ^a^	22 ± 0.2 ^a^	25 ± 0.3 ^a^	29 ± 0.1 ^a^	30 ± 0.4 ^a^	32 ± 0.3 ^a^	35 ± 0.3 ^a^
*C. glabrata*	−	17 ± 0.5 ^c^	19 ± 0.2 ^c^	21 ± 0.9 ^c^	23 ± 0.6 ^d^	25 ± 0.6 ^d^	26 ± 0.2 ^c^
*C. albicans*	13 ± 0.1 ^c^	20 ± 0.7 ^b^	22 ± 0.5 ^b^	25 ± 0.5 ^bc^	27 ± 0.4 ^bc^	29 ± 0.9 ^bc^	31 ± 0.0 ^b^
*C. davenportii*	15 ± 0.3 ^b^	21 ± 0.1 ^ab^	24 ± 0.8 ^ab^	27 ± 0.2 ^b^	28 ± 0.3 ^b^	30 ± 0.1 ^b^	31 ± 0.1 ^b^

Data are presented as mean ± SD, lowercase letters (a, b, c, d, …) in the same columns indicate significant differences (effect of each pp concentration on pathogenic bacteria and fungi) *p* ≤ 0.05. * St, penicillin (1000 µg/mL) for bacteria and clotrimazole (1000 µg/mL) for fungi.

**Table 5 bioengineering-09-00735-t005:** The minimum concentration of PP extract that inhibits or kills bacteria or fungi.

Pathogenic Bacteria	MIC *	MBC *
*S. aureus*	50 ^f^	90 ^f^
*L. monocytogenes*	70 ^d^	130 ^d^
*B. cereus*	65 ^e^	120 ^e^
*P. aeruginosa*	110 ^b^	190 ^b^
*K. pneumonia*	150 ^a^	250 ^a^
*E. coli*	90 ^c^	150 ^c^
Pathogenic fungi	MIC	MFC *
*A. niger*	120 ^b^	200 ^b^
*A. flavus*	55 ^d^	90 ^e^
*P. expansum*	45 ^e^	80 ^f^
*C. glabrata*	160 ^a^	290 ^a^
*C. albicans*	75 ^c^	130 ^c^
*C. davenportii*	70 ^cd^	120 ^d^

Lowercase letters (a, b, c, d, …) in the same columns indicate significant differences (effect of each pp concentration on pathogenic bacteria and fungi) *p* ≤ 0.05. * MIC, minimum inhibitory concentration; * MBC, minimum bactericidal concentration; * MFC, minimum fungicidal concentration of PP extract against tested pathogens.

**Table 6 bioengineering-09-00735-t006:** Changes in physiochemical parameters of smoothie supplemented with pomegranate pomace extract at different addition levels (0.4, 0.8, 1.2, 1.6 mg/mL) during two-month storage at 4 °C.

Smoothie Samples	Storage Period (Months)	pH	Acidity (mg/10 mL)	Vit. C (mg/mL)	Total Sugars (mg/mL)	Fat (%)	TSS (%)
SYSC	0	4.98 ± 0.4 ^a^	55.32 ± 0.1 ^d^	6.3 ± 0.1 ^a^	1.20 ± 0.01 ^a^	1.10 ± 0.2 ^g^	9.30 ± 0.2 ^e^
	1	4.75 ± 0.2 ^b^	58.78 ± 0.2 ^d^	4.0 ± 0.2 ^cd^	0.91 ± 0.02 ^cd^	1.30 ± 0.4 ^f^	10.10 ± 0.1 ^de^
	2	4.69 ± 0.3 ^c^	61.22 ± 0.3 ^cd^	2.7 ± 0.2 ^e^	0.78 ± 0.01 ^e^	1.50 ± 0.7 ^e^	10.90 ± 0.3 ^d^
PP-SYS1	1	4.68 ± 0.2 ^c^	62.39 ± 0.7 ^c^	4.2 ± 0.0 ^c^	0.92 ± 0.09 ^cd^	1.53 ± 0.5 ^e^	10.90 ± 0.95 ^d^
	2	4.60 ± 0.0 ^c^	64.72 ± 0.4 ^bc^	3.0 ± 0.1 ^de^	0.81 ± 0.03 ^d^	1.60 ± 0.2 ^de^	11.60 ± 0.1 ^c^
PP-SYS2	1	4.55 ± 0.1 ^d^	62.59 ± 0.9 ^c^	4.4 ± 0.3 ^c^	0.93 ± 0.05 ^c^	1.54 ± 0.1 ^e^	11.02 ± 0.7 ^cd^
	2	4.43 ± 0.3 ^ef^	65.00 ± 0.1 ^b^	3.5 ± 0.5 ^d^	0.89 ± 0.06 ^d^	1.72 ± 0.3 ^c^	12.03 ± 0.0 ^b^
PP-SYS3	1	4.51 ± 0.4 ^d^	62.99 ± 0.3 ^c^	4.8 ± 0.8 ^c^	0.95 ± 0.04 ^c^	1.62 ± 0.4 ^de^	11.72 ± 0.0 ^c^
	2	4.44 ± 0.6 ^ef^	66.05 ± 0.7 ^b^	3.8 ± 0.6 ^d^	0.92 ± 0.01 ^cd^	1.79 ± 0.7 ^b^	12.55 ± 0.2 ^ab^
PP-SYS4	1	4.49 ± 0.1 ^e^	63.20 ± 0.5 ^c^	5.2 ± 0.3 ^b^	1.09 ± 0.00 ^b^	1.69 ± 0.3 ^d^	11.88 ± 0.1 ^c^
	2	4.40 ± 0.5 ^f^	67.02 ± 0.2 ^a^	4.2 ± 0.1 ^c^	1.00 ± 0.03 ^bc^	1.82 ± 0.2 ^a^	12.89 ± 0.2 ^a^

Data are presented as mean ± SD, lowercase letters (a, b, c, d, …) in the same columns indicate significant differences (between the changes in physiochemical parameters of SYS samples during 2-month storage period) at *p* ≤ 0.05.

**Table 7 bioengineering-09-00735-t007:** Fluctuations in sensory traits of smoothie supplemented with pomegranate pomace extract at different addition levels (0.4, 0.8, 1.2, 1.6 mg/mL) during two months of storage at 4 °C.

Smoothie Samples	Storage Period (Months)	Flavor	Color	Texture	Over Acceptability
SYSC	0	8.4 ± 0.1 ^bc^	8.3 ± 0.3 ^d^	8.4 ± 0.5 ^b^	8.3 ± 0.2 ^c^
	1	8.1 ± 0.3 ^c^	8.0 ± 0.2 ^e^	8.0 ± 0.5 ^c^	8.0 ± 0.1 ^cd^
	2	7.5 ± 0.2 ^d^	7.7 ± 0.1 ^f^	7.5 ± 0.3 ^d^	7.6 ± 0.3 ^d^
PP-SYS1	1	8.6 ± 0.0 ^b^	8.5 ± 0.2 ^cd^	8.4 ± 0.4 ^b^	8.5 ± 0.4 ^bc^
	2	8.2 ± 0.3 ^c^	8.3 ± 0.2 ^d^	7.9 ± 0.6 ^d^	8.1 ± 0.3 ^cd^
PP-SYS2	1	8.8 ± 0.1 ^b^	8.6 ± 0.0 ^c^	8.4 ± 0.7 ^b^	8.6 ± 0.2 ^b^
	2	8.4 ± 0.2 ^bc^	8.4 ± 0.0 ^cd^	8.0 ± 0.3 ^c^	8.2 ± 0.4 ^c^
PP-SYS3	1	8.9 ± 0.3 ^ab^	8.8 ± 0.2 ^b^	8.5 ± 0.4 ^b^	8.7 ± 0.3 ^b^
	2	8.5 ± 0.5 ^bc^	8.5 ± 0.1 ^cd^	8.0 ± 0.6 ^c^	8.3 ± 0.2 ^c^
PP-SYS4	1	9.0 ± 0.0 ^a^	9.0 ± 0.0 ^a^	8.8 ± 0.1 ^a^	8.9 ± 0.3 ^a^
	2	8.6 ± 0.3 ^b^	8.8 ± 0.3 ^b^	8.5 ± 0.2 ^b^	8.6 ± 0.4 ^b^

Data are presented as mean ± SD. Lowercase letters in the same columns indicate significant differences *p* ≤ 0.05. Smoothie supplemented with pomegranate pomace extract 0.4 mg/mL (PP-SYS1), smoothie supplemented with pomegranate pomace extract 0.8 mg/mL (PP-SYS2), smoothie supplemented with pomegranate pomace extract 1.2 mg/mL (PP-SYS3), smoothie supplemented with pomegranate pomace extract 1.6 mg/mL (PP-SYS4).

## Data Availability

Not applicable.

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
