# Peer review of "Pomegranate Pomace Extract with Antioxidant, Anticancer, Antimicrobial, and Antiviral Activity Enhances the Quality of Strawberry-Yogurt Smoothie"

_bioengineering, 2022, doi:10.3390/bioengineering9120735_

Round 1

Reviewer 1 Report

This work has scientific merit and should be considered for publication. However, I suggest the acceptance only after major alterations in language and grammar. In addition, the results presented in the tables could be divided to improve the understanding of the statistical differences, which are confusing in this set up. The discussion should be seriously improved.

Author Response

Reviewer 1 comments

Comments and Suggestions for Authors

This work has scientific merit and should be considered for publication. However, I suggest the acceptance only after major alterations in language and grammar. In addition, the results presented in the tables could be divided to improve the understanding of the statistical differences, which are confusing in this set up. The discussion should be seriously improved.

Response: Thanks for the reviewer for the positive comments on our manuscript. We carefully responded to all comments that highlighted in yellow color

The linguistic mistakes were revised by an expert and highlighted with purple color. Some tables were separated, and significant letters were checked for clarity in all tables and figures. Additionally, the discussion was enhanced.

Reviewer 2 Report

The manuscript entitled “Pomegranate Pomace Extract with Antioxidant, Anticancer, Antimicrobial, and Antiviral Activity; Its Effects on Preserving and Supporting the Quality of Strawberry-Yogurt Smoothie” can be accepted after addressing the following suggested minor comments.

1.      Title needs to revised in concise scientific style.

2.      A single line motivation needs to introduced in abstract.

3.      All graph and figures quality must be improved.

4.      Figure 3 A, scale bar is missing from the micrograph.

5.      In antimicrobial activity, what authors suggestion for employing higher concentrations.

6.      Antimicrobial activity, add up the Petri plats in supplementary data or replace the data with quantitative micro plate method of activity.

7.      For broader readership add up the following suggested references:

https://doi.org/10.3389/fonc.2021.741326

https://doi.org/10.3390/molecules26154593

Author Response

Reviewer 2 comments

Comments and Suggestions for Authors

The manuscript entitled “Pomegranate Pomace Extract with Antioxidant, Anticancer, Antimicrobial, and Antiviral Activity; Its Effects on Preserving and Supporting the Quality of Strawberry-Yogurt Smoothie” can be accepted after addressing the following suggested minor comments.

Response: Thanks for the reviewer for his efforts in enhancing the manuscript. All comments have considered accordingly.

  1. Title needs to revised in concise scientific style.

Response: The title was reformulated accordingly

  1. A single line motivation needs to introduced in abstract.

Response: It was added in the beginning of abstract

  1. All graph and figures quality must be improved.

Response: The quality of figures and graphs were enhanced accordingly

  1. Figure 3 A, scale bar is missing from the micrograph.

Response: The scale bar was added

  1. In antimicrobial activity, what authors suggestion for employing higher concentrations.

Response: we can initiate a prediction model with equations to predict the antimicrobial of higher concentration of PP

  1. Antimicrobial activity, add up the Petri plats in supplementary data or replace the data with quantitative micro plate method of activity.

Response: The available plates image was added in the supplementary

  1. For broader readership add up the following suggested references:

https://doi.org/10.3389/fonc.2021.741326

https://doi.org/10.3390/molecules26154593

Response: The suggested references were cited in the discussion of anticancer of PP

Reviewer 3 Report

Pomegranates are one of the fruits commodities that have high productivity, but about 60% of its fruit is pomace which is often discarded as waste. Pomegranate pomace contains bioactive compounds such as phenolic compounds. Therefore, an exploration of the usage of pomegranate pomace as a food additive to strawberry yogurt smoothies to improve the quality of the product is needed. The author’s study would be an excellent finding for further utilization of pomegranate pomace. However, it requires several improvements before it can be considered for publication.

Abstract: 

- what are WAC and OAC?

- "Using pomegranate pomace as a source of polyphenols and fiber in functional foods enhances SYS's physiochemical and sensory qualities." - Pomegranate pomace as a source of polyphenols and fiber can enhance SYS's physiochemical and sensory qualities.

- Add information about antioxidant activity result

Keywords:

- please use other keywords that are different from the manuscript title to enhance discoverability

Introduction:

- line 52: 1,500,000 tons worldwide per annum?

- line 59: anthocyanidins are a part of flavonoids too

- line 71: should be written 0.20-1.0 mg/mL

- line 79: what are PPE and MIC?

Materials and method:

- line 141: centrifuge unit should be written with consistent unit

- inconsistent words: min or minutes

- line 183: "emission" should not be written in bold format

- table 1: the sentence "data are presented as mean± SD" should not be written below the table due to the table just give information about the formula and ingredients.

- what kind of panelist that used in the sensory testing?

Result and Discussion:

- line 285: "Table 1 shows that the OHC of PP is 7.77 mL/g" table 1 only provides information about the formula, not the OHC value

- "50" in SC50 and IC50 should be written with subscript format

- Tables 4 and 5: Please add the information about the usage of the notation in the table

Conclusion:

- which formula that has the best performance as a food additive for SYS products?

Overall:

- Temperature unit should be written separately from the value ( 4 ˚C)

- Inconsistent words: ml or mL?

- the significant figures should be written consistently  

- mass unit should be written separated with the value.

Author Response

Reviewer 3 comments

Comments and Suggestions for Authors

Pomegranates are one of the fruits commodities that have high productivity, but about 60% of its fruit is pomace which is often discarded as waste. Pomegranate pomace contains bioactive compounds such as phenolic compounds. Therefore, an exploration of the usage of pomegranate pomace as a food additive to strawberry yogurt smoothies to improve the quality of the product is needed. The author’s study would be an excellent finding for further utilization of pomegranate pomace. However, it requires several improvements before it can be considered for publication.

Response: Thanks for the reviewer. All comments have considered

Abstract: 

- what are WAC and OAC?

Response: Water absorption capacity (WAC) and Oil absorption capacity (OAC), and they detailed in respected place

- "Using pomegranate pomace as a source of polyphenols and fiber in functional foods enhances SYS's physiochemical and sensory qualities." - Pomegranate pomace as a source of polyphenols and fiber can enhance SYS's physiochemical and sensory qualities.

- Add information about antioxidant activity result

 Response: Thanks for this comment. We added some information about antioxidant activity in abstract

Keywords:

- please use other keywords that are different from the manuscript title to enhance discoverability

 Response: Done as requested

Introduction:

- line 52: 1,500,000 tons worldwide per annum?

 Response: Thanks for the reviewer, yes, per year, it was added in the text

- line 59: anthocyanidins are a part of flavonoids too

 Response: It was adjusted accordingly

- line 71: should be written 0.20-1.0 mg/mL

 Response: Done as requested

- line 79: what are PPE and MIC?

  Response: It was detailed in the text

Materials and method:

- line 141: centrifuge unit should be written with consistent unit

 Response: Done as requested

- inconsistent words: min or minutes

 Response: The units were unified through the manuscript

- line 183: "emission" should not be written in bold format

 Response: Done as requested

- table 1: the sentence "data are presented as mean± SD" should not be written below the table due to the table just give information about the formula and ingredients.

 Response: Thanks for the reviewer. It was a typing mistake and it was deleted

- what kind of panelist that used in the sensory testing?

  Response: 30 semi-trained panelists

Result and Discussion:

- line 285: "Table 1 shows that the OHC of PP is 7.77 mL/g" table 1 only provides information about the formula, not the OHC value

 Response: It was adjusted accordingly

- "50" in SC50 and IC50 should be written with subscript format

Response: Done as requested

- Tables 4 and 5: Please add the information about the usage of the notation in the table

  Response: The notation in the tables were clarified accordingly

Conclusion:

- which formula that has the best performance as a food additive for SYS products?

  Response: The addition of PP extract (1.6 mg/mL) to SYS samples considerably enhances color, texture, and quality and introduce functional beverage may be beneficial for many patients.

Overall:

- Temperature unit should be written separately from the value (4 ˚C)

Response: Done as requested

- Inconsistent words: ml or mL?

 Response: It was unified as mL

- the significant figures should be written consistently  

 Response: Done as requested

- mass unit should be written separated with the value.

Response: Done as requested